# Association of Plasma Carotenoid and Malondialdehyde Levels with Physical Performance in Korean Adolescents

**DOI:** 10.3390/ijerph19074296

**Published:** 2022-04-03

**Authors:** Heeyeon Joo, Jiyoung Hwang, Ji Yeon Kim, Saejong Park, Hyesook Kim, Oran Kwon

**Affiliations:** 1Graduate Program in System Health Science and Engineering, Department of Nutritional Science and Food Management, Ewha Womans University, Seoul 03760, Korea; joohy1127@naver.com (H.J.); cindy.jyhwang@gmail.com (J.H.); 2Department of Food Science and Technology, Seoul National University of Science and Technology, Seoul 01811, Korea; jiyeonk@seoultech.ac.kr; 3Department of Sport Science, Korea Institute of Sport Science, Seoul 01794, Korea; saejpark@kspo.or.kr

**Keywords:** malondialdehyde (MDA), carotenoid, physical performance, antioxidant, adolescent, effect modifier

## Abstract

Increased oxidative stress has been shown to lead to muscle damage and reduced physical performance. The antioxidant mechanism is most likely to reduce these relationships, but in the context of the action of carotenoids, more research is needed. This study aimed to investigate whether carotenoids modify the association between plasma malondialdehyde (MDA) and physical performance in Korean adolescents. The study sample consisted of 381 adolescents (164 boys, 217 girls) aged 13–18, who participated in the 2018 National Fitness Award Project. We quantified α-carotene, β-carotene, β-cryptoxanthin, lutein, zeaxanthin, lycopene, and MDA levels in plasma using HPLC with photodiode array detection. Among boys but not girls, plasma MDA level was negatively associated (*β* = −0.279, *p* = 0.0030) with total plasma carotenoid levels and marginally negatively associated (*β* = −0.907, *p* = 0.0876) with absolute hand grip strength. After adjustment for covariates in boys, the MDA level was negatively associated with absolute hand grip strength and relative hand grip strength; this association was observed only in groups with individual carotenoid and total carotenoid values below the median. These findings support a significant association between plasma MDA level and hand grip strength, and this association has been potentially modified by plasma levels of carotenoids in Korean male adolescents.

## 1. Introduction

Physical performance (or fitness) is a measure of physical activity that requires cardiorespiratory endurance, neuromuscular skeletal endurance and strength, and flexibility [1,2]. An individual’s physical performance state is mainly determined by lifestyle factors, including physical activity and genetically inherited ability. Oxidative stress status is considered one of the most important factors related to physical performance [3,4].

Malondialdehyde (MDA) is one of the most common biomarkers of oxidative stress as an end product of lipid peroxidation [5]. A number of clinical [6,7] and epidemiological [8,9] studies have evaluated the degree of oxidative stress/oxidative damage by measuring MDA content in humans. A recent study indicated that an elevated MDA level was associated with a higher risk of low hand grip strength in the non-paretic limbs of stroke patients [10].

Carotenoids are natural pigments responsible for the yellow, orange, and red colors of various fruits and vegetables [11]. Human organisms cannot synthesize carotenoids; therefore, these compounds must be supplied with the diet. Among the various carotenoids supplied through the diet, α-carotene, β-carotene, β-cryptoxanthin, lutein, zeaxanthin, and lycopene are the six main carotenoids found in human blood serum [12]. Carotenoid status has been associated with various health benefits. Findings have linked the importance of carotenoids to human health with their antioxidant activity, protecting cells and tissues from oxidative damage [13]. Carotenoids are very potent scavengers of reactive oxygen species (ROS), and their absorption and storage in the human body are superior to other phytochemicals (e.g., flavonoids) [14].

Some observational evidence suggests that carotenoids are positively associated with physical performance [15,16,17]. Our previous study observed that plasma carotenoid levels might be positively associated with physical performance in Korean adolescents [18]. It could be that the protective effect of carotenoids against oxidative damage promotes health and nutrition to elevate physical performance. To the best of our knowledge, no current literature studies have investigated the association between blood carotenoids, MDA, and physical performance.

There have also been few studies examining such associations during adolescence, a critical period considering that the physical performance level in adolescence can affect diseases in adulthood. Therefore, the aim of this study is to determine whether carotenoid levels modify the association between the blood MDA level and physical performance in Korean adolescents aged 13–18 years.

## 2. Materials and Methods

### 2.1. Study Participants

The subjects were adolescents who participated in the Korean National Fitness Award Project in 2018. This large-scale national project is currently managed by 21 centers [19]. Among the 450 adolescents in our previous study, which reported associations between carotenoids and physical performance [18], we excluded participants for whom we were unable to analyze MDA concentration due to insufficient blood samples (*n* = 69). As a result, 381 participants (164 boys, 217 girls) aged 13–18 years were included in this study. This study was approved by the Institutional Review Board (IRB) of the Korea Institute of Sport Science, and Ewha Womans University. Informed consent of all participants was obtained before enrollment.

### 2.2. General Characteristics and Anthropometric Measurements

Skilled interviewers surveyed the study participants’ smoking, drinking, physical activity, and eating habits to obtain general information on their health-related behaviors. We defined “current smoker” as a participant who reported smoking at least one cigarette in the last 30 days and “current drinker” as a participant who reported drinking at least one cup of alcohol in the last 30 days. We defined “physical activity” as being involved in at least one of four intensity levels (high, moderate, cardio-intensive, and strength-intensive) at least once a week. Questions related to eating habits consisted of breakfast frequency in the last week, fruits, fast foods, carbonated drink consumption, and the number of late-night snacks in the last 30 days.

Height was measured in units of 0.1 cm using a stadiometer (Seca, Seca Corp., Columbia, MD, USA). Body weight was measured in units of 0.1 kg using an electronic weight scale (Inbody 720, Biospace, Seoul, Korea). Body mass index (BMI) was calculated as weight divided by height squared (kg/m^2^). All parameters were measured by skilled medical staff.

### 2.3. Plasma MDA Measurements

We used an approved high-performance liquid chromatography (HPLC) method to quantify plasma MDA based on its reaction with 2-thiobarbituric acid (TBA). The HPLC instrument (Shiseido Co., Ltd., Tokyo, Japan) was equipped with an analytical column (4.6 mm × 250 mm; Shiseido Co., Ltd.) for MDA separation and a fluorescence detector (excitation length = 527 nm, emission length = 551 nm) for MDA detection. The mobile phase consisted of 50 mM potassium phosphate buffer (pH 6.8) and methanol (7:3, *v/v*). The flow rate was 1.0 mL/min at 40 °C. To calibrate the peak of the MDA–TBA adduct, a 1,1,3,3-tetra-ethoxypropane solution was used [20].

Before the analysis, plasma samples were mixed with 0.44 M phosphoric acid and a 42 mM phosphoric acid solution for deproteinization. Then, they were heated at 95 °C for 1 h, cooled at 4 °C for 1 h, and centrifuged (2500× *g* at 4 °C for 3 min). The resultant supernatants were filtered through a 0.45-μm PTFE syringe filter.

### 2.4. Plasma Carotenoids Measurements

Blood samples were taken after participants fasted for 8 h. Plasma was obtained by immediate centrifugation of the blood samples at 3000 rpm for 3 min. Plasma carotenoid levels were determined on an HPLC instrument (Shiseido Co., Ltd.) equipped with a YMC C30 column (5 µm, 4.6 × 250 mm; YMC Europe GmbH, Dinslaken, Germany) and a photodiode array detector (Shiseido Co., Ltd.). We quantified α-carotene, β-carotene, β-cryptoxanthin, lutein, zeaxanthin, and lycopene (Appendix A). Total carotenoids were calculated as the sum of the six individual carotenoids.

### 2.5. Physical Performance Measurements

Physical performance was assessed by determining muscular strength (absolute and relative hand grip strength test), muscular endurance (curl-up test), and aerobic capacity (20-m Progressive Aerobic Cardiovascular Endurance Run: 20-m PACER test, estimated maximal oxygen consumption: VO_2max_). The National Fitness Award project test items for Korean adolescents showed a high consistency, with reliability ranging from 0.87 to 0.99 [21]. All physical performance assessments were conducted by trained experts, and the detailed evaluation methods are as follows:Muscular strength

To assess the absolute hand grip strength (kg, %), the participants first extended both feet shoulder-width apart in an upright position. A hand dynamometer (GRIP-D 5101, Takei, Niigata, Japan) was adjusted to the second finger of the participants. Then, the arms of the participant were straightened down and kept 15° apart from the torso. At the signal “start,” the participant exerted maximum strength to hold the hand dynamometer for 5 s. The maximum value was recorded to the nearest 0.1 kg for the left then right hand. The relative hand grip strength (kg, %) was calculated using the formula: (absolute hand grip strength (kg)/body weight (kg)) × 100.
Muscular endurance

The curl-up test (number of times) was performed to assess muscular endurance. First, the participant was instructed to lay down with their knees bent, and their feet fixed to the floor about 30 cm away from their hips. At the signal “start,” the participant extended their arms forward until their fingertips touched their knees, then returned to the starting position. This movement was regarded as one curl-up. Each participant repeated this process. The number of curl-ups was measured and recorded.
Aerobic capacity

To perform the 20 m PACER test (number of times), the experts divided each lane into a 20 m course and drew a line with tape at both ends. Participants were required to run the 20 m back and forth across the marked track, keeping time with beeps. The “start” beep was signaled 5 s after the “ready” command, at which point the participants started to run the 20 m. If they reached the relevant line before the beep, they had to wait until the second beep before running toward the end of the opposite line. If participants did not reach the relevant line before the first beep, they would run in the opposite direction at the second beep. They were eliminated if they could not reach the relevant line before the second beep. In this way, participants must continue until the line is not reached before the second beep, and the eliminator must stay clear of the line. The maximum number of repetitions was recorded.

Another effective aerobic capacity measurement index is maximal oxygen uptake, VO_2max_. However, it is costly and hard to measure VO_2max_ directly. Instead, VO_2max_ was estimated by the 20 m PACER using the quadratic model developed by Mahar et al. [22] to estimate VO_2max_ (mL/kg·min) in adolescents.

### 2.6. Statistical Analysis

General characteristics were presented as mean ± standard deviation for continuous variables and as numbers and percentages for categorical variables. The differences in the mean general characteristics, plasma carotenoid levels, plasma MDA levels, and physical performances between the groups below the median and above the median were analyzed using Student’s *t*-test. Differences in the distribution of categorical variables, such as smoking, drinking, and physical activity, between the groups below the median and above the median were analyzed by the chi-square test. To determine whether the carotenoid modified the association between plasma MDA levels and physical performances, multiple linear regression analysis was used after adjusting for age, BMI, smoking, drinking, and physical activity. In the case of relative hand grip strength, body weight was already applied in the calculation process, so it was analyzed by excluding BMI from these covariates. All analyses were performed using SAS software (version 9.4; SAS Institute, Inc., Cary, NC, USA). Significance was defined as a value of *p* < 0.05.

## 3. Results

### 3.1. General Characteristics, Plasma Carotenoid Level, Plasma MDA Level, and Physical Performances

A total of 381 participants (164 boys, 217 girls) aged 13–18 years were recruited for this cross-sectional study. As shown in Table 1, age, height, weight, BMI, percentage of physically active participants, and percentage of current smokers were significantly higher in boys than girls. There was no significant difference in the eating habits between boys and girls (Appendix A).

The mean level of β-cryptoxanthin was significantly higher (*p* = 0.0213) in girls than in boys, whereas the mean level of zeaxanthin was higher (*p* = 0.0206) in boys than in girls (Table 2). The mean levels of all physical performance were significantly higher (*p* < 0.0001 for all) in boys than in girls (Table 3).

### 3.2. Associations between Plasma Carotenoid and MDA Levels and Physical Performance

Table 4 and Table 5 show the multiple linear regression analysis results of the associations between plasma carotenoid level, plasma MDA level, and physical performance after adjusting for age, BMI, smoking, drinking, and physical activity. Plasma MDA level was marginally negatively associated (*β* = −0.907, *p* = 0.0876) with absolute hand grip strength in boys but not in girls (Table 4). In boys, the levels of β-carotene (*p* = 0.0007), β-cryptoxanthin (*p* < 0.0001), lutein (*p* = 0.0003), zeaxanthin (*p* = 0.0139), and total carotenoids (*p* = 0.0030) in plasma were negatively associated with the plasma MDA level. In girls, the levels of β-carotene (*p* = 0.0426), β-cryptoxanthin (*p* = 0.0011), and total carotenoids (*p* = 0.0867, marginally) in plasma were negatively associated with the plasma MDA level (Table 5).

As shown in Table 6, in boys, the negative associations between plasma MDA level and absolute handgrip strength were observed only in groups with α-carotene (*β* = −1.497, *p* = 0.0153), β-cryptoxanthin (*β* = −1.829, *p* = 0.0280), zeaxanthin (*β* = −1.841, *p* = 0.0164), and total carotenoid (*β* = −1.825, *p* = 0.0424) values below the median. In addition, the plasma MDA level was negatively associated with relative hand grip strength in groups with β-carotene (*β* = −3.473, *p* = 0.0199), β-cryptoxanthin (*β* = −3.786, *p* = 0.0066), zeaxanthin (*β* = −2.809, *p* = 0.0301), and total carotenoid (*β* = −4.011, *p* = 0.0077) values below the median. No significant associations between plasma MDA level and physical performances according to the carotenoid median were observed in girls.

## 4. Discussion

To our knowledge, there are no previous studies examining the association between plasma MDA level and hand grip strength in adolescents. In this study, we found that plasma MDA level was marginally negatively associated with hand grip strength in Korean male adolescents. This finding is consistent with recently reported data showing associations between an elevated plasma MDA level and an increased risk of low hand grip strength in post-stroke patients [10]. Another previous study revealed a negative correlation between MDA and the musculoskeletal index, and MDA has been accepted as a potential early biomarker of sarcopenia in older adults [23]. The reason that the association between MDA and hand grip strength was marginally rather than strong in our study is thought to be because our participants were general adolescents who did not have much-decreased muscle strength. The results of our study are meaningful in that they suggest the possibility that MDA may be related to grip strength in general adolescents, not older adults or patients.

In this study, we found that individual carotenoids, including β-carotene, and total carotenoid levels were negatively correlated with MDA. Several randomized controlled trials examining the effect of carotenoid supplementation on the reduction of MDA levels [24,25,26,27,28] focused mainly on β-carotene [26,28], which is known to have high antioxidant activity. Similar to our findings in a cross-sectional study, Sayyah reported a negative relationship between MDA and β-carotene in 50 asthmatic patients in Basrah, Iraq [29]. To our knowledge, only one other study has examined the association between the β-carotene level and MDA level in adolescents in age ranges similar to our participants. That work observed no association between plasma β-carotene deficiency and MDA level in Spanish schoolchildren aged 9–12 [30]. These different results are probably related to differences in race, ethnicity, and food culture, and more large-scale studies with healthy adolescents are needed in the future.

We also found that the MDA level was negatively associated with absolute hand grip strength and relative hand grip strength; this association was observed only in groups with individual carotenoid and total carotenoid values below the median in Korean male adolescents. These results suggest that the association between MDA and hand grip strength was potentially modified by the plasma carotenoid levels. Although it is difficult to explain the exact mechanism for the results of this study, the role of carotenoids as effect modifiers in the association between MDA and hand grip strength could be linked to the role of carotenoids as part of the antioxidant defense system in humans. High levels of oxidative stress in the blood indicate the presence of many ROS, which can cause muscle damage [31,32,33] and eventually lead to physical performance loss [34,35]. Carotenoids are well known to be effective lipophilic antioxidants that quench singlet oxygen [36] and scavenge other ROS [37]. Studies of older adults yielded evidence to support the positive association between carotenoid levels and physical performance [38,39]. Moreover, a previous study on Korean adolescents observed a positive relationship between plasma carotenoid levels and physical performance [18].

In addition to total carotenoids, only α-carotene, β-carotene, β-cryptoxanthin, and zeaxanthin, among the six individual carotenoids studied, acted as effect modifiers in the association between MDA and hand grip strength. We do not know why there was no significance for lutein and lycopene. One possible explanation is that the bioavailability of carotenoids does not depend solely on physiological mechanisms but can be affected by many other factors, such as gender [40], dietary factors [41], and health status [42]. In particular, the natural food matrix or the microstructure of processed foods is a major dietary factor [41]. As one of the most lipophilic carotenoids (along with β-carotene), it is perhaps not surprising that lycopene bioavailability may vary depending on the concurrent ingestion of dietary fat [43]. Differences in the lipophilicity and number of polar groups between carotenoids affect their partitioning in the gut and thus their availability to the intestinal micelles for transport to absorptive sites. Interactions among carotenoids may hinder carotenoid bioavailability, and carotenoids retained in an unabsorbable fat-soluble phase also show decreased adsorption [41]. Moreover, Borel et al. [44] argued that differences in genetic polymorphism were related to lycopene absorption and metabolism.

In the current study, hand grip strength was the only significant physical performance measure associated with the MDA level. Hand grip strength is the most used indicator of muscle strength [45] and has been used as a representative indicator of physical fitness [46]. Matsudo et al. [47] suggested that hand grip strength can act as a very accurate and independent predictor of physical fitness in both children and adolescents. Similarly, Gerodimos [48] suggested the reliability of hand grip strength in basketball players from childhood to adulthood.

We do not know exactly why the association between MDA and hand grip strength by carotenoids median is common in boys. The bioactivity of carotenoids is affected by various physiological factors in men and women. A study of Europeans argued that women had higher blood carotenoid levels than men and that gender differences existed [49]. Similarly, in a study of older adults in Europe, factors related to the status of circulating carotenoid levels were investigated, and the total carotenoids were especially higher in women than in men [50]. Thus, the concentration of circulating carotenoids is influenced by gender, and further studies of the concentration and bioactivity of carotenoids in men are needed.

Our study has several limitations. First, because of its cross-sectional design, we cannot determine the causal relationship between carotenoid and MDA with physical performance in our study. Second, there may be unmeasured variables that result in unmeasured (or residual) confounding (e.g., genetic factors, protein intake, vitamin C or E intake, sleep status). Especially, we measured only the plasma level of carotenoids, an important variable in this study, but not the dietary intake, including carotenoid-containing supplements. However, blood carotenoids are usually considered a better measure of data, taking into account all consumption as well as bioavailability in the body rather than nutrient intakes (if the carotenoids have a long half-life and are repeatedly measured to represent long-term intake) [51]. Irrespective of individual variations in absorption, availability, and metabolism [52,53], plasma concentrations of carotenoids mirror the intake of fruits and vegetables because of their abundance in these foods [52]. Previous research has shown a dose–response association between intake and appearance of carotenoids in plasma [54], making carotenoids a fairly reliable biomarker of total carotenoid intake. Third, the sample size was relatively small. However, despite these limitations, to the best of our knowledge, this study is the first to suggest that the association between oxidative stress status and physical performance in Korean adolescents may depend on the level of carotenoids in the plasma. Considering that the physical performance level in adolescence can affect diseases in adulthood, our findings are clinically meaningful.

## 5. Conclusions

In conclusion, we found that MDA level was negatively associated with hand grip strength, and the plasma carotenoid levels potentially modified this association in Korean male adolescents. These findings suggest the possibility of partly shifting the association between MDA and hand grip strength by dietary carotenoid consumption. Intervention studies and larger sample sizes are needed to develop our results about the association of plasma carotenoid and MDA levels with physical performance.

## Figures and Tables

**Table 1 ijerph-19-04296-t001:** General characteristics of Korean adolescents ^1^.

	Total (*n* = 381)	Boys (*n* = 164)	Girls (*n* = 217)	*p*
General characteristics				
Age (years)	15.3 ± 2.0	15.5 ± 2.0	15.0 ± 2.0	0.0162
Height (cm)	164.9 ± 7.6	170.0 ± 7.1	161.0 ± 5.4	<0.0001
Weight (kg)	57.5 ± 11.8	62.8 ± 12.5	53.5 ± 9.5	<0.0001
BMI (kg/m^2^)	21.0 ± 3.5	21.6 ± 3.8	20.6 ± 3.1	0.0032
Physical activity (*n*, %)	326 (85.6)	152 (92.7)	174 (80.2)	0.0006
Current smokers (*n*, %)	11 (2.9)	8 (4.9)	3 (1.4)	0.0436
Current drinkers (*n*, %)	38 (10.0)	20 (12.2)	18 (8.3)	0.2084

^1^ Values are expressed as mean ± standard deviation or *n* (%). Current smokers indicate a participant who reported to have smoked at least one cigarette in the last 30 days. Current drinkers indicate a participant who reported drinking at least one cup of alcohol in the last 30 days. Physical activity indicates participation in at least one of the four intensities (high, moderate, cardio-intensive, and strength-intensive) at least once a week. BMI, body mass index.

**Table 2 ijerph-19-04296-t002:** Plasma carotenoid and MDA concentrations of Korean adolescents ^1^.

	Total (*n* = 381)	Boys (*n* = 164)	Girls (*n* = 217)	*p*
Plasma carotenoid concentration (μmol/L)				
α-Carotene	0.16 ± 0.05	0.16 ± 0.06	0.16 ± 0.05	0.4111
β-Carotene	0.58 ± 0.34	0.56 ± 0.39	0.59 ± 0.30	0.3627
β-Cryptoxanthin	0.44 ± 0.25	0.41 ± 0.23	0.47 ± 0.26	0.0213
Lutein	0.25 ± 0.10	0.24 ± 0.09	0.25 ± 0.11	0.0907
Zeaxanthin	0.20 ± 0.07	0.20 ± 0.07	0.19 ± 0.06	0.0206
Lycopene	0.55 ± 0.31	0.56 ± 0.28	0.54 ± 0.33	0.4226
Total carotenoids ^2^	2.17 ± 0.69	2.13 ± 0.79	2.20 ± 0.61	0.3859
Plasma MDA concentration (μmol/L)	3.0 ± 1.1	2.9 ± 1.0	3.0 ± 1.2	0.3365

^1^ Values are expressed as mean ± standard deviation. ^2^ Total carotenoids: the sum of the levels of the six individual carotenoids in plasma. MDA, malondialdehyde.

**Table 3 ijerph-19-04296-t003:** Physical performance levels of Korean adolescents ^1^.

	Total (*n* = 381)	Boys (*n* = 164)	Girls (*n* = 217)	*p*
Physical Performance				
Absolute handgrip strength (kg)	28.8 ± 8.8	36.0 ± 7.7	23.4 ± 4.9	<0.0001
Relative handgrip strength (%)	50.3 ± 12.5	58.2 ± 11.4	44.4 ± 9.7	<0.0001
20 m PACER (reps)	36.0 ± 16.7	47.9 ± 15.9	27.1 ± 10.6	<0.0001
Estimated VO_2max_ (mL/kg·min)	39.6 ± 5.5	44.4 ± 4.2	35.9 ± 3.0	<0.0001
Curl-up (reps)	25.9 ± 17.4	32.8 ± 17.6	20.6 ± 15.4	<0.0001

^1^ Values are expressed as mean ± standard deviation. PACER, progressive aerobic cardiovascular endurance run; VO_2max_, maximal oxygen uptake.

**Table 4 ijerph-19-04296-t004:** Multiple linear regression analysis for the association between plasma MDA levels and physical performance in Korean adolescents ^1^.

	Absolute Handgrip Strength			Relative Handgrip Strength			20-m PACER			Estimated VO_2max_			Curl-Up		
	*β*	SE	*p*	*β*	SE	*p*	*β*	SE	*p*	*β*	SE	*p*	*β*	SE	*p*
Boys (*n* = 164)															
Plasma MDA	−0.907	0.528	0.0876	−1.369	0.921	0.1391	−0.181	1.278	0.8879	−0.035	0.251	0.8879	−1.172	1.407	0.4062
Girls (*n* = 217)															
Plasma MDA	0.018	0.271	0.9479	−0.560	0.571	0.3279	−0.404	0.602	0.5022	−0.079	0.118	0.5022	−0.267	0.908	0.7686

^1^ Adjusted for age, BMI (except for relative handgrip strength), smoking, drinking, and physical activity. BMI, body mass index; SE, standard error; MDA, malondialdehyde; PACER, progressive aerobic cardiovascular endurance run; VO_2max_, maximal oxygen uptake.

**Table 5 ijerph-19-04296-t005:** Multiple linear regression analysis for the association between plasma carotenoid and MDA levels in Korean adolescents ^1^.

	Plasma MDA		
	*β*	SE	*p*
Boys (*n* = 164)			
α-Carotene	3.407	1.343	0.0622
β-Carotene	−0.628	0.183	0.0007
β-Cryptoxanthin	−1.275	0.301	<0.0001
Lutein	−2.962	0.793	0.0003
Zeaxanthin	−2.603	1.046	0.0139
Lycopene	0.295	0.265	0.2685
Total carotenoids ^2^	−0.279	0.093	0.0030
Girls (*n* = 217)			
α-Carotene	2.618	1.556	0.0623
β-Carotene	−0.544	0.267	0.0426
β-Cryptoxanthin	−1.015	0.308	0.0011
Lutein	−0.354	0.757	0.6408
Zeaxanthin	1.506	1.249	0.2294
Lycopene	0.106	0.242	0.6602
Total carotenoids ^2^	−0.224	0.130	0.0867

^1^ Adjusted for age, BMI (except for relative handgrip strength), smoking, drinking, and physical activity. BMI, body mass index; SE, standard error; MDA, malondialdehyde. ^2^ Total carotenoids: the sum of the level of the six individual carotenoids in plasma.

**Table 6 ijerph-19-04296-t006:** Multiple linear regression analysis for the association between plasma MDA level and physical performance in Korean adolescents by the carotenoid median ^1^.

			Absolute Handgrip Strength			Relative Handgrip Strength			20-m PACER			Estimated VO_2max_			Curl-Up		
			*β*	SE	*p*	*β*	SE	*p*	*β*	SE	*p*	*β*	SE	*p*	*β*	SE	*p*
Boys (*n* = 164)	Median																
*α*-Carotene	0.194	Below the median	**−1.497**	0.603	**0.0153**	−1.875	1.004	0.0656	−1.247	1.377	0.3677	−0.245	0.270	0.3677	−2.404	1.523	0.1187
		Above the median	0.585	0.968	0.5475	0.360	1.869	0.8477	1.380	2.571	0.5931	0.270	0.504	0.5931	1.766	2.847	0.5371
*β*-Carotene	0.465	Below the median	−1.321	0.920	0.1553	**−3.473**	1.460	**0.0199**	0.476	1.991	0.8117	0.093	0.390	0.8117	−0.578	2.202	0.7937
		Above the median	−0.341	0.648	0.6002	0.404	1.118	0.7191	−0.369	1.703	0.8292	−0.072	0.334	0.8292	−1.626	1.949	0.4068
*β*-Cryptoxanthin	0.347	Below the median	**−1.829**	0.816	**0.0280**	**−3.786**	1.355	**0.0066**	0.363	1.886	0.8479	0.071	0.370	0.8479	−1.800	1.928	0.3536
		Above the median	−0.274	0.698	0.6956	0.576	1.176	0.6258	−0.157	1.791	0.9305	−0.031	0.351	0.9305	−0.130	2.096	0.9506
Lutein	0.239	Below the median	−1.378	0.745	0.0682	−1.911	1.267	0.1357	−0.697	1.896	0.7143	−0.137	0.372	0.7143	0.380	1.892	0.8415
		Above the median	−0.176	0.854	0.8375	−0.129	1.516	0.9324	1.233	1.993	0.5380	0.242	0.391	0.5380	−1.154	2.383	0.6297
Zeaxanthin	0.225	Below the median	**−1.841**	0.750	**0.0164**	**−2.809**	1.271	**0.0301**	−0.722	1.658	0.6642	−0.142	0.325	0.6642	−0.841	1.684	0.6190
		Above the median	0.146	0.809	0.8573	0.627	1.484	0.6740	1.427	2.238	0.5257	0.280	0.439	0.5257	−1.264	2.538	0.6200
Lycopene	0.547	Below the median	−1.406	0.912	0.1273	−2.996	1.678	0.0781	0.433	2.127	0.8392	0.085	0.417	0.8392	−2.047	2.380	0.3925
		Above the median	−0.659	0.630	0.2992	−0.766	1.034	0.4610	−0.684	1.612	0.6726	−0.134	0.316	0.6726	−0.228	1.806	0.9001
Total carotenoids ^2^	1.962	Below the median	**−1.825**	0.884	**0.0424**	**−4.011**	1.464	**0.0077**	1.077	1.839	0.5599	0.211	0.360	0.5599	−1.038	1.895	0.5853
		Above the median	−0.250	0.662	0.7063	0.643	1.086	0.5560	−0.476	1.793	0.7912	−0.093	0.351	0.7912	−0.466	2.170	0.8305
Girls (*n* = 217)	Median																
*α*-Carotene	0.139	Below the median	−0.411	0.359	0.2549	−0.799	0.675	0.2391	−0.420	1.100	0.7037	−0.082	0.216	0.7037	−1.802	1.423	0.2082
		Above the median	0.228	0.419	0.5863	−0.171	0.922	0.8536	−0.483	0.729	0.5095	−0.095	0.143	0.5095	0.155	1.271	0.9030
*β*-Carotene	0.510	Below the median	0.160	0.361	0.6573	−0.250	0.718	0.7284	−0.798	0.752	0.2911	−0.156	0.147	0.2911	0.534	0.948	0.5746
		Above the median	−0.071	0.418	0.8651	−0.886	0.927	0.3415	0.129	0.949	0.8918	0.025	0.186	0.8918	−0.967	1.584	0.5429
*β*-Cryptoxanthin	0.388	Below the median	0.331	0.446	0.4590	−0.377	0.940	0.6889	−0.429	0.875	0.6246	−0.084	0.171	0.6246	0.422	1.492	0.7777
		Above the median	−0.083	0.368	0.8227	−0.604	0.785	0.4431	−0.153	0.889	0.8639	−0.030	0.174	0.8639	−0.446	1.195	0.7100
Lutein	0.240	Below the median	−0.537	0.371	0.1511	−0.538	0.683	0.1265	−1.067	1.066	0.3190	−0.209	0.209	0.3190	−0.399	1.393	0.7752
		Above the median	0.359	0.403	0.3749	0.080	0.887	0.9279	0.055	0.746	0.9417	0.011	0.146	0.9417	0.261	1.220	0.8311
Zeaxanthin	0.188	Below the median	−0.568	0.410	0.1695	−1.000	0.791	0.2094	−0.081	1.146	0.9437	−0.016	0.225	0.9437	−0.443	1.584	0.7805
		Above the median	0.275	0.380	0.4713	−0.221	0.837	0.7920	−0.485	0.708	0.4949	−0.095	0.139	0.4949	−0.218	1.166	0.8521
Lycopene	0.516	Below the median	0.386	0.381	0.3133	−0.152	0.764	0.8430	−0.307	0.814	0.7065	−0.060	0.159	0.7065	0.502	1.268	0.6931
		Above the median	−0.581	0.397	0.1467	−1.250	0.905	0.1701	−0.515	0.924	0.5787	−0.101	0.181	0.5787	−1.589	1.350	0.2418
Total carotenoids ^2^	2.092	Below the median	0.081	0.373	0.8293	−0.623	0.727	0.3930	−0.697	0.852	0.4153	−0.137	0.167	0.4153	1.135	1.197	0.3455
		Above the median	0.060	0.414	0.8854	−0.541	0.921	0.5585	−0.318	0.877	0.7177	−0.062	0.172	0.7177	−1.497	1.404	0.2891

^1^ Adjusted for age, BMI (except for relative handgrip strength), smoking, drinking, and physical activity; BMI, body mass index; SE, standard error; MDA, malondialdehyde; PACER, progressive aerobic cardiovascular endurance run; VO_2max_, maximal oxygen uptake. ^2^ Total carotenoids: the sum of the levels of the six individual carotenoids in plasma.

## Data Availability

The dataset used and/or analyzed during this study is available from the corresponding author upon reasonable request.

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
