# Peer review of "Association of Plasma Carotenoid and Malondialdehyde Levels with Physical Performance in Korean Adolescents"

_ijerph, 2022, doi:10.3390/ijerph19074296_

Round 1

Reviewer 1 Report

This article reports how plasma carotenoids such as alpha-carotene, beta-carotene, beta-cryptoxanthin, zeaxanthin, lutein, and lycopene associate with Malondialdehyde levels and physical strength of Korean adolescents. The authors clearly explain the methods and the limitations of their study. 
Since beta-carotene and some other carotenoids can convert into retinoids, how is the retinoid concentration? Will this affect the carotenoid concentration at the end?
Also, carotenoid content in plasms depends on the diet, so did any participants mention taking carotenoid-containing supplements?
If possible, it's worth showing a representative HPLC elution profile for carotenoid separation.

Author Response

This article reports how plasma carotenoids such as alpha-carotene, beta-carotene, beta-cryptoxanthin, zeaxanthin, lutein, and lycopene associate with Malondialdehyde levels and physical strength of Korean adolescents. The authors clearly explain the methods and the limitations of their study.

Reply) We sincerely appreciate the reviewer's insightful and constructive comments and suggestions. Please see our detailed responses below.

  1. Since beta-carotene and some other carotenoids can convert into retinoids, how is the retinoid concentration? Will this affect the carotenoid concentration at the end?

Reply) We did not analyze blood retinoid concentrations in this study. Therefore, it is not possible to know whether retinoid levels are related to carotenoid levels. Next time we have the opportunity, we would like to consider this.

  1. Also, carotenoid content in plasma depends on the diet, so did any participants mention taking carotenoid-containing supplements?

Reply) We did not investigate whether any of the participants took carotenoid supplements. We included this limitation to our study methodology in the Discussion section (L301-310).

  1. If possible, it’s worth showing a representative HPLC elution profile for carotenoid separation.

Reply) As suggested, we included the HPLC elution profile for carotenoid separation in Supplementary Fig. S1 (L108).

Reviewer 2 Report

The paper deals with "Association of Plasma Carotenoid and Malondialdehyde Levels with Physical Performance in Korean Adolescents".  The objective is interesting and fits with the scope of the Journal. In addition is a trendy topic. The introduction is well-written and the tables are well presented, although some little details can be improved. I marked them into the text as a comment.

Author Response

The paper deals with "Association of Plasma Carotenoid and Malondialdehyde Levels with Physical Performance in Korean Adolescents". The objective is interesting and fits with the scope of the Journal. In addition is a trendy topic. The introduction is well-written and the tables are well presented, although some little details can be improved. I marked them into the text as a comment.

Reply) We sincerely appreciate the reviewer's constructive comments.

  1. It would be better for the readers understanding, if the author divides this table into three parts and include the data for generla chara. in table 1, carotenoid and MDA levels in table 2 and physical activities in table 3.

Reply) Thank you for the valuable comments. As suggested, we have divided Table 1 into three parts: general characteristics in Table 1, carotenoid and MDA concentrations in Table 2, and physical performances in Table 3 for better understanding.

  1. What are these values for? is this value and the values below are the total number of participant as in the above column? a bit confusing.

Reply) As suggested above, we have revised Table 1 to reduce the confusion.

Reviewer 3 Report

Thanks for submitting the manuscript “Association of Plasma Carotenoid and Malondialdehyde Levels with Physical Performance in Korean Adolescents” to Int. J. Environ. Res. Public Health. Although the subject of the study is interesting, I believe that questions need to be answered before any decision is made.

Introduction

This session should provide information on why carotenoids were the bioactive molecule chosen and not another compound eg flavonoids. My suggestion is to reframe this session with that in mind.

Methods & Results

Line#77-79: were these results displayed? My concern in this work is that data from the diet of the participants, such as average carotenoid intake, were not demonstrated. If they were investigated they should at least appear as a supplementary file. This is because the carotenoids in the plasma may be due to ingestion as after digestion and absorption it is transported by this route to the storage tissue or it may be due to the use of the stored compound by the body's need. But not having the diet data can lead to a false positive or negative correlation depending on the participant's intake. This doubt causes statements such as line 226 to be questioned: is the decrease in the amount of carotenoids due to an increase in MDA or is the decrease in the amount of carotenoids due to low intake?

Author Response

Thanks for submitting the manuscript “Association of Plasma Carotenoid and Malondialdehyde Levels with Physical Performance in Korean Adolescents” to Int. J. Environ. Res. Public Health. Although the subject of the study is interesting, I believe that questions need to be answered before any decision is made.

  1. Introduction. This session should provide information on why carotenoids were the bioactive molecule chosen and not another compound eg flavonoids. My suggestion is to reframe this session with that in mind.

Reply) Thank you for the meaningful suggestion. As recommended, we added a sentence explaining why we chose carotenoids and not other compounds, such as flavonoids, in the Introduction section (L48-50).

  1. Methods & Results. Line#77-79: were these results displayed? My concern in this work is that data from the diet of the participants, such as average carotenoid intake, were not demonstrated. If they were investigated they should at least appear as a supplementary file. This is because the carotenoids in the plasma may be due to ingestion as after digestion and absorption it is transported by this route to the storage tissue or it may be due to the use of the stored compound by the body's need. But not having the diet data can lead to a false positive or negative correlation depending on the participant's intake. This doubt causes statements such as line 226 to be questioned: is the decrease in the amount of carotenoids due to an increase in MDA or is the decrease in the amount of carotenoids due to low intake?

Reply) As suggested, the results of eating habits (L77-79 in the original manuscript) were presented in Supplementary Table S1 and additionally described in the Results section (L170-171). It would be best if dietary carotenoid intake was also measured along with blood carotenoid levels, as the reviewers commented, but we did not investigate dietary carotenoid intake in this study. We have clearly stated these limitations to our study methodology. Due to these limitations, as highlighted in your comment, we found a negative association between plasma carotenoid levels and MDA levels in this study, but we may not know exactly whether a decrease in carotenoid levels is due to an increase in MDA or a decrease in carotenoid intake. However, we added a reference to show that carotenoid in plasma is a fairly reliable biomarker of carotenoid intake, and we further elaborated on this (L301-310).

Round 2

Reviewer 3 Report

The authors improve the manuscript a lot and can be accept now.